# A Bit Stickier, a Bit Slower, a Lot Stiffer: Specific vs. Nonspecific Binding of Gal4 to DNA

**DOI:** 10.3390/ijms22083813

**Published:** 2021-04-07

**Authors:** Thomas Carzaniga, Giuliano Zanchetta, Elisa Frezza, Luca Casiraghi, Luka Vanjur, Giovanni Nava, Giovanni Tagliabue, Giorgio Dieci, Marco Buscaglia, Tommaso Bellini

**Affiliations:** 1Dipartimento di Biotecnologie Mediche e Medicina Traslazionale, Università di Milano, 20054 Segrate (MI), Italy; thomas.carzaniga@unimi.it (T.C.); lucasiraghi@gmail.com (L.C.); luka.vanjur@unimi.it (L.V.); giovanni.nava@unimi.it (G.N.); 2CiTCoM, CNRS, Université de Paris, F-75006 Paris, France; elisa.frezza@parisdescartes.fr; 3Proxentia S.r.l., Viale Ortles 22/4, 20139 Milano, Italy; giovanni.tagliabue@proxentia.com; 4Dipartimento di Scienze Chimiche, della Vita e della Sostenibilità Ambientale, Università di Parma, 43124 Parma, Italy; giorgio.dieci@unipr.it

**Keywords:** transcription factors, biosensor, protein–DNA interactions, molecular dynamics, biophysics, entropy–enthalpy compensation

## Abstract

Transcription factors regulate gene activity by binding specific regions of genomic DNA thanks to a subtle interplay of specific and nonspecific interactions that is challenging to quantify. Here, we exploit Reflective Phantom Interface (RPI), a label-free biosensor based on optical reflectivity, to investigate the binding of the N-terminal domain of Gal4, a well-known gene regulator, to double-stranded DNA fragments containing or not its consensus sequence. The analysis of RPI-binding curves provides interaction strength and kinetics and their dependence on temperature and ionic strength. We found that the binding of Gal4 to its cognate site is stronger, as expected, but also markedly slower. We performed a combined analysis of specific and nonspecific binding—equilibrium and kinetics—by means of a simple model based on nested potential wells and found that the free energy gap between specific and nonspecific binding is of the order of one kcal/mol only. We investigated the origin of such a small value by performing all-atom molecular dynamics simulations of Gal4–DNA interactions. We found a strong enthalpy–entropy compensation, by which the binding of Gal4 to its cognate sequence entails a DNA bending and a striking conformational freezing, which could be instrumental in the biological function of Gal4.

## 1. Introduction

Protein–DNA interactions play essential roles in several biological functions in cells, like gene transcription, DNA replication, repair and recombination. To perform their regulatory functions, many of the DNA-binding proteins, among which are transcription factors (TF), need to bind to specific double-stranded (ds) sites in the presence of an overwhelming number of nonspecific dsDNA tracts. These proteins thus have to be optimized both for specific binding and for effective searching, which proceeds through a combination of sliding along the dsDNA, switching between contacting dsDNA segments and hopping via unbinding, 3D diffusion and binding [1,2,3]. Therefore, the specific binding, its strength and molecular conformations, are just one of the key ingredients for the regulation mechanisms. Equally crucial are the kinetic on and off rates—that gauge the role of hopping, the strength and nature of the nonspecific binding — that control the sliding and the flexibility, and multivalence of the protein—that lower the barrier for intersegment switching. The balance among these many factors is often achieved through the combination of two or more positively charged DNA-binding domains and the presence of unstructured protein tracts. Sequence-specific recognition is generally anticipated by nonspecific interactions provided by electrostatics and by the formation of some hydrogen bonds (HB) between protein residues and DNA. Specific interactions with the cognate site are obtained through additional HB, which require a precise mutual positioning of the protein and dsDNA. This process involves an enthalpy gain but, also, an entropy penalty associated with the reduction of the available molecular conformations [4,5].

The simultaneous optimization of all these requirements is very difficult to achieve, as suggested by the fact that, in the presence of large genomes, such as in eukaryotic cells, the selectivity of gene regulators is not provided by single DNA-binding proteins but, rather, is attained through the cooperation of various transcription factors, each of which is only partially selective for its cognate site. In this context, a complete investigation of the binding process of proteins targeting a specific site in the genome should take into account not only the strength and selectivity of the cognate site but, also, the nature and strength of the nonspecific binding, the kinetic parameters and the interactions with the cofactors.

Historically, the quantitative evaluation of DNA binding by transcription factors has first focused on identifying the cognate sites and the strength and selectivity of the specific binding. This is, by itself, a challenging task, since the binding landscape of this class of proteins is known to be particularly complex, much beyond the simple view of one single consensus sequence with high specificity for one single protein [6,7,8,9], as also evidenced by state-of-the-art computer simulations [10]. The knowledge of specific protein–DNA interactions also includes various detailed structural descriptions of crystallized protein–DNA complexes that have become available in recent years. The same approach, however, cannot be extended to nonspecific interactions that remain more elusive.

Given the synergic contribution of specific and nonspecific bindings, relevant new understanding could emerge from their simultaneous measurement and combined analysis. Not many of these studies are currently available, reflecting the fact that the available experimental approaches are all, to some degree, inadequate to determine the equilibrium and kinetics of both strong and weak binding processes. Indeed, the difference between specific and nonspecific binding has hardly any solid quantitative reference: reported estimates for the specific/nonspecific free energy differences range from 1 to 7 kcal/mol [11,12], reflecting both the intrinsic differences in behaviors between proteins and the spread in approaches and sensitivity of the experimental methods. Electrophoretic mobility shift assays (EMSA) and protein arrays are the two most used methods to determine specific but, also, nonspecific interactions of DNA-binding proteins. By requiring hour-long bond lifetimes, the duration of the electrophoretic run, EMSA is suitable for strong, stable binding but not as much for weaker or more dynamic binding, such as those required to provide the necessary interplay between hopping and sliding in the search process [13]. Protein-binding microarrays enable comparing the binding at the equilibrium of a given protein to a large variety of DNA strands [14]. The conversion of a fluorescence signal into an equilibrium dissociation constant is, however, not straightforward, losing the accuracy for weak binding, which is also extremely sensitive to image processing and subtraction of the background signal [15] (see Appendix A). Alternative methods include label-free biosensing techniques, which have demonstrated enough sensitivity and access to kinetic behavior [16] but were never focused on comparative specific/nonspecific protein–DNA binding.

The Reflective Phantom Interface (RPI) is an optical, label-free technology based on reflectivity, which has been introduced by our group and applied to protein–protein [17,18] and DNA–DNA interactions [19,20]. Since the RPI enables the real-time, multi-spot detection of the molecular binding, it allows to investigate in parallel the kinetics and equilibrium of the interaction of a protein with different specific and nonspecific dsDNA fragments.

Here, we applied the RPI to the study of the paradigmatic gene regulator Gal4 of Saccharomyces cerevisiae [21,22], a sequence-specific DNA-binding protein with a zinc finger-type DNA-binding domain. Gal4 activates the transcription of various galactose-inducible genes by binding to a specific upstream activation sequence (UASG), thereby favoring the recruitment of the RNA polymerase II transcription machinery to a downstream-located core promoter region [23]. As schematically shown in Figure 1A, Gal4 binds DNA as a homodimer, with two zinc finger domains (Zn2/Cys6-fold group) making base pair-specific contacts to highly conserved CGG triplets at the ends of the consensus sequence, while flexible linkers and dimerization elements contact the phosphate backbone within the inner 11 base pairs. We selected Gal4 because of a combination of factors: it has been extensively studied with different approaches, including label-free biosensing [24], and a crystallographic description of its interaction with the cognate site is available [21]; despite thorough structural knowledge, the mechanisms yielding selective regulation by this and other yeast transcription factors are still unclear [25,26].

Considered together, the previous investigations of Gal4–DNA binding, which took place in the last decades and involved a variety of techniques, give a sense of the intrinsic uncertainty of the experimental determinations in this field: the available estimates for the dissociation constant of its complex with the consensus sequence span several orders of magnitude, ranging from 200 nM [24] down to 0.5 nM [27]. Most studies, however, found values in the 3–30-nM range [22,26,28], in line with several other transcription factors [29,30]. The available data for the kinetics of Gal4 unbinding from its cognate site give *k_off_* ≈ 10^−4^ s^−1^, obtained with single-molecule experiments [31] and Surface Plasmon Resonance imaging [24]. Nonspecific interactions measured by EMSA indicate 10- to 1000-fold weaker binding [26], in line with what was found for other zinc finger proteins [15].

In this work, we present a thorough study of specific and nonspecific Gal4–dsDNA binding, which includes methodological novelty, the developing of a simple kinetic model to analyze the results and performing MD simulations to interpret the results. Specifically, we exploited the RPI to measure the equilibrium and kinetics of Gal4–dsDNA binding at different ionic strengths. We perform a combined analysis of specific and nonspecific binding using a hierarchical two-step process model, which enables extracting the difference in free energy between the two modes of interaction. In parallel, we performed long, state-of-the-art all-atom molecular dynamics simulations of Gal4 contacting dsDNA, which offered a detailed description of the specific and nonspecific binding of Gal4, including protein conformations, bond distribution and DNA bending. Experimental data and computer simulations consistently indicate that the binding of Gal4 to its cognate site involves a marked entropy/enthalpy compensation.

## 2. Results

### 2.1. Gal4 Binding to Specific and Nonspecific DNA Sequences

The sensing surface of the RPI is a nonreflecting glass substrate, coated with a polymer to reduce nonspecific binding, on which “receptors” are chemically immobilized in spots [18,19] (Figure 1B). The RPI raw signal is the reflected light intensity from each spot on the sensing surface, which can be converted, with no free parameter, into the molecular surface mass density *σ* (see Appendix A). To explore specific vs. nonspecific Gal4–DNA binding, we prepared surfaces with spots containing four distinct dsDNA probes: two 40 basepair (bp) blunt-ended dsDNA and two hairpins (40 bp-long ds stem plus eight-base-long loop), differing only for the presence or absence of the consensus sequence from position 28 to position 45 (Figure 1B and Appendix A). All probes have a single-strand spacer of 10 adenosines to provide flexibility and increase the distance to the surface. The consensus sequence we used, 5′ CGG-N_11_-CCG 3′, is based on a large body of previous studies showing sequence-specific DNA binding by Gal4 as a dimer (e.g., Marmorstein [21]; see Appendix A). The high conservation of CGG motifs has been recently confirmed by protein-binding microarray (PBM) studies to reflect a high affinity for the Gal4 DNA-binding domain of each monomer [32]. In this study, we considered as nonspecific any sequence lacking such a consensus and analyzed in detail two of them, the NSP sequence (Figure 1B) and the CTRL sequence (Appendix A and Appendix A), the latter chosen to minimize an affinity for cryptic sites (see Appendix A for more details).

Examples of Gal4-binding curves measured for spots of specific and nonspecific dsDNA probes (listed in Table 1) are shown in Figure 1B. When the consensus sequence is present, the amount of bound proteins is larger and the time to saturation longer, indicating a stronger but overall slower interaction with the specific tract. This behavior can be better appreciated in experiments in which Gal4 is added in stepwise increasing concentrations *c*. Figure 2A shows *σ(t)*, the time evolution of the protein mass accumulating on the four families of spots following the injection of Gal4 in the measuring cell. At all concentrations, the binding of Gal4 to the spots carrying the consensus sequence is more “efficient”, since these spots capture a larger amount of Gal4 proteins, and “slower”, since it takes more time to plateau. A similar difference is also observed with respect to other nonspecific control strands with different sequences (Appendix A).

We have fitted each increment of *σ*(*t*) by
(1)$$\sigma \left(t\right)\;=\;\Sigma \left(c\right)\left(1-e^{-\Gamma \left(c\right)t}\right)$$
where the two fitting parameters are the extrapolated asymptote at each injected concentration *Σ(c)* and the growth rate Γ(c). It is worth noticing that, since $\Gamma \left(c\right)\;=\;k_{on}c+k_{off}$ and *k_on_* and *k_off_* are the kinetic rates for binding and unbinding, the measurement of the rising time does not simply reflect the binding rate but, rather, conveys information on both.

Figure 2B shows the values obtained for the asymptotic value *Σ(c)* for the pair of specific and nonspecific hairpin probes. These can, in turn, be fitted with a simple Langmuir adsorption curve:(2)$$\Sigma \left(c\right)\;=\;\Sigma_{\infty }/\left(1+\frac{K_{d}}{c}\right)$$
where the dissociation constant *K_d_* = *k_off_*/*k_on_*. In the fitting process, the saturation value *Σ**_∞_* is not constrained but is kept the same for each pair of specific and nonspecific probes. This corresponds to the assumption that the maximum number of proteins hosted by a single probe duplex at large protein concentration depends on the probe length but not on the presence of the specific tract. Remarkably, the value of *Σ**_∞_* obtained from repeated measurements in the same conditions of Figure 2 corresponds to 0.9 ± 0.1 Gal4 homodimers per DNA strand (see Appendix A). This evidence does not exclude the possibility that at large concentration more than one protein can bind to a single DNA probe strand, either containing the specific tract or not. Indeed, the total length of the dsDNA probes roughly corresponds to the size of two Gal4 homodimers. However, our analysis indicates that the possible binding of a second protein on the same DNA strand is either unlikely or characterized by a much larger *K_d_*, hence not affecting the analysis of the initial part of the Langmuir isotherm proposed in this study.

This analysis enables determining the *K_d_* summarized in Figure 2D. As expected, Gal4 interacts with its cognate site more strongly (*K_d_* = 25–35 nM) than with generic sequences (*K_d_* = 160–240 nM). These values indicate a free-energy difference of about 0.9-1.2 kcal/mol between specific and nonspecific sequence, similar to what was observed for the Max protein in reference [11]. The values only slightly depend on the dsDNA probe density on the spots (Appendix A) and on the background treatment (Appendix A). The association rate *k_on_* is obtained from the measured initial slope *σ**’(c)* after each stepwise concentration increment (Figure 2C). Since $\sigma^{\prime }\left(c\right)\;=\;\Sigma \left(c\right)\Gamma \left(c\right)=\;\Sigma_{\infty }k_{on}c$, *k_on_* is obtained as the slope of the linear fit of *σ’(c)/**Σ**_∞_*. The extracted *k_on_* is very similar for specific and nonspecific interactions, being less than 20% larger in specific spots, suggesting an equality to the *k_on_*. By adopting this assumption, i.e., fitting all data as a single set (dashed line in Figure 2C), we obtain *k_on_* = 1.6 ± 0.6 × 10^−5^ s^−1^ nM^−1^, with the differences in *K_d_* mainly ascribed to the different rate *k_off_* = *K_d_*·*k_on_*, with which Gal4 unbinds from duplexes. In the case of strands carrying the Gal4 consensus sequence, we obtain *k_off_* = 3.1–6.9 × 10^−4^ s^−1^, while, in the case of generic dsDNA, *k_off_* = 1.4–4.7 × 10^−3^ s^−1^, indicating a detachment time almost 10 times faster. The measured *k_off_* for the consensus sequence is similar within a factor of three to the value obtained for Gal4 in previous studies by Surface Plasmon Resonance imaging [24], whereas our value of *k_on_* is about 25 times larger, suggesting a faster access of the protein to the DNA strands on the surface in our conditions. Besides differences in the composition and passivation of the sensing surface, it must be noted that our approach relies on a global analysis of both specific and nonspecific binding (see Appendix A) measured at low concentrations of Gal4, thus far from the saturation of surface probe sites and consequent crowding effects. In general, similar equilibrium or kinetic rate constants of protein interactions can be determined with both solution-phase methodologies and surface biosensors [33]. However, the surface immobilization of nucleic acids can provide an accumulation of charges that induces the electrostatic effects typical of large solution-phase concentrations comparable to those of the DNA within the nucleus [20].

To better explore the relevance of an electrostatic component of the interactions, we performed experiments at various values of ionic strength *I_s_* across the standard value *I_s_* = 150-mM NaCl. The electrostatic effects are relevant, as indicated by the decreasing bound proteins (Appendix A). The behaviors of *K_d_*, *k_on_* and *k_off_* vs. *I_s_* are shown in Figure 3A–C. In presence of specific bonds, *K_d_* progressively increases with *I_s_* (Figure 3A, red dots). A similar behavior has been observed for various DNA-binding proteins and is mostly related to the release of counterions upon binding [34,35]. In the absence of specific interactions, *K_d_* increases with *I_s_* more rapidly, indicating a stronger weakening of the nonspecific bonds up to *I_s_* = 200-mM NaCl, above which it sharply falls (Figure 3A, blue dots). This nonmonotonic behavior leads to a maximum difference between specific and nonspecific equilibrium constants, remarkably located around 150-mM NaCl. Further insight is given by the kinetics. *k_on_* monotonically decreases with I_s_, as expected from the reduced electrostatic attraction range (Figure 3C).

The escape rate of Gal4 from a generic dsDNA is made easier by increasing the salt concentration up to 150-mM NaCl, above which *k_off_* sharply drops (Figure 3B). When specific interactions are present, *k_off_* is instead monotonic and much milder. This somewhat surprising behavior could be understood in the following way. The weakening of the electrostatic attraction is more relevant for nonspecific interactions, which are less stabilized by HB. However, at large *I_s_*, the value of nonspecific *k_off_* approaches that of specific interactions, indicating a similar stability in the two situations and thus suggesting that the narrowed electrostatic self-repulsion favors the onset of new attractive interactions, possibly additional HB made accessible by previously inaccessible conformations.

### 2.2. Analysis of Equilibrium and Kinetics through a Nested–Well Binding Model

The specific docking of Gal4 to its consensus sequence is known to depend on the formation of about 20 HB (Figure 1A), which require Gal4 to be in a precise position and orientation with respect to the dsDNA and to adopt a definite molecular conformation. Thus, when Gal4 is in contact with its consensus sequence but its position/orientation/conformation is not the one enabling H-bonding, its interaction must resemble those relevant for generic dsDNA. This agrees with the notion that interactions of Gal4 to its consensus sequence are intrinsically preceded by those to nonspecific dsDNA that control sliding and hopping [36]. Since we have parallel access to specific and nonspecific observations, we aim to disentangle the two components by performing a differential analysis of our data.

To this goal, we developed a simple model embodying this notion of specific-through-nonspecific interactions. Our model shares features with previous ones that were proposed to incorporate into simple kinetic equations the notion that the target search of transcription factor crucially depends on nonspecific binding, which might have the role of an “antenna” [37] or of a “funnel” [38], facilitating the docking on the cognate site. The model we propose here is, however, simpler than previous ones, as a consequence of the simultaneous access, afforded by our experimental design, to the binding to specific and nonspecific DNA strands of equal sizes.

In our model, we introduce a reaction coordinate x, ordering all possible Gal4 molecular conformations, which are the same around the DNA strands that carry or not the specific sequence. We thus envision the specific binding as encompassing a set of x coordinates surrounded by regions of nonspecific interactions, as in the “Nested–Well” (NW) model sketched in Figure 1C and discussed in Appendix A, which comprises two consecutive binding reactions: reaction 1 (from unbound to nonspecific interactions) and reaction 2 (from nonspecific interactions to specific binding). In the latter, the equilibrium dissociation coefficient *K*_2_ is defined as *K*_2_ = *σ*_1_/*σ*_2_, the ratio between the surface densities of nonspecifically (*σ*_1_) and specifically bound (*σ*_2_) proteins to the dsDNA probes. In the limit of large *K*_2_—i.e., vanishing depth of the inner well—the NW model becomes a single-well model describing the binding to strands that do not carry the target sequence, with the equilibrium coefficient *K*_1_. The binding equilibrium that is measured when Gal4 interacts with dsDNA carrying its cognate site, and involves both specific docking and nonspecific interactions, should instead be compared to the binding of the whole NW system. In this case, *K_d_* = *K*_1_*K*_2_/(1+ *K*_2_), or
(3)$$K_{2}\;=\;\frac{K_{d}}{K_{1}-K_{d}}$$

Having experimental access to both *K*_1_ and *K_d_* (see Figure 3A) it is straightforward to extract *K*_2_ as a function of the ionic strength, as shown in Figure 3D. We find that the smallest *K*_2_, corresponding to the tightest specific binding, is found at the concentration of 150-mM NaCl.

The solution of the NW model also provides the time dependence of the amount of Gal4 adhering to DNA after a stepwise increment of its concentration, which is found to depend on *k_on_*_1_ and *k_off_*_1_—the kinetic rate constants for the nonspecific binding of Gal4 on the duplexes and *k_on_*_2_ and *k_off_*_2_—the rate constants for the unimolecular reaction from the nonspecific to the specific state for the proteins already bound to the DNA. We find that the binding kinetics to a NW probe is a double exponential, with shorter (*τ_S_*) and longer (*τ_L_*) characteristic times:(4)$$\sigma \left(c,t\right)\;=\;\Sigma \left(c\right)\;\left(1-B\;e^{-t/\tau_{L}}-\left(1-B\right)\;e^{-t/\tau_{S}}\right)$$
where *B*, *τ_S_* and *τ_L_* are explicit functions of the equilibrium and kinetic coefficients for transitions 1 and 2 (see Appendix A). At moderate *K*_2_, of interest for this analysis, the response time is nearly exponential and is dominated by *τ_L_* (*B* ≈ 1). *τ_L_* depends on *k_off_*_2_: when *k_off_*_2_ is small, *τ_L_* becomes large, as expected because of the slower escape time from the inner well; when *k_off_*_2_ is large, τ_L_ reaches a limiting value larger than the response time for nonspecific binding ($\tau_{L\infty }\;$> τ_1_). $\tau_{L\infty }$ depends on *K*_1_, *K*_2_ and *τ*_1_ and corresponds to the time involved in the escape from the outer well of the nonspecifically bound proteins that, however, are in constant equilibrium with those that are specifically bound (Appendix A). The data shown in Figure 1B were simultaneously fit to this kinetic model (continuous lines). We find a good agreement with the data, indicating that the NW model captures the differences in both the binding strength and kinetics. In particular, we find $\tau_{L}\approx$
$\tau_{L\infty }$, as apparent by comparing the NW fitting curve with such limiting exponential behavior $\sigma \left(c,t\right)\;=\;\Sigma \left(c\right)\;\left(1-\;e^{-t/\tau_{L\infty }}\right)$ (dashed line), indicating that the residence time of Gal4 on its consensus sequence is shorter than *k_off_*_1_^−1^ ≈ 300 s. The nearly exponential kinetics, predicted and observed, justifies the use of Equation (1) in the general analysis of our data. Despite its simplicity, the NW model enables describing the Gal4 binding by capturing the slower approach to the saturation of specific interactions by justifying that *k_on_* is the same in specific and nonspecific interactions and by providing a mean to quantify the nonspecific-to-specific transition. Further details are available in Appendix A.

### 2.3. Entropy–Enthalpy Compensation

The selectivity for Gal4 to its consensus sequence observed in this study and expressed by the coefficient *K*_2_ is, at best, $K_{2}\approx 0.16$ at 150-mM NaCl, meaning that, out of 10 Gal4 dimers bound to the dsDNA probe containing the consensus sequence, 10/1.16 ≈ 8.6 are actually docked to a cognate sequence, while 1.4 contact the dsDNA without adopting the specific binding conformation. One could wonder how this 6:1 ratio could manage to regulate the gene expression in vivo, where the ratio between the number of DNA bases involved in the consensus sequence vs. all bases present in the system is not of the order of 10^−1^, as it is here, but, rather, of the order 10^−6^ or less, a notion suggesting that only a tiny minority of the Gal4 molecules actually manages to dock on the DNA target.

The weak selectivity revealed by *K*_2_ indicates that the free energy difference between specific and nonspecific binding is rather small, $\Delta G\;=\;RT\ln\left(K_{2}\right)\approx -1.1\;\mathrm{kcal}/\mathrm{mol}\;$, where *R* is the gas constant. Intriguingly, this figure is much smaller than the one expected from the large number of HB involved in the docking of Gal4, which should provide an enthalpic gain upon specific binding an order of magnitude larger [39,40,41]. To explore the entropic and enthalpic components of ΔG, we performed binding experiments analogous to those in Figure 2 as a function of the temperature (see Appendix A and Appendix A) from which we extracted *K*_2_*(T)*, shown in Figure 4.

By fitting these data with $K_{2}=exp\left(\Delta H/RT-\Delta S/R\right)$ (dashed line), we obtain ΔH≈−12.8±3 kcal/mol  and ΔS≈−38.7±9cal/(mol K), confirming the expectation of an enthalpic gain upon specific binding more than 10 times larger than the measured ΔG, which is compensated for more than 90% by a similarly large entropic penalty. This emerging entropy–enthalpy compensation indicates that the enthalpy made available by the HB is spent more in entropy reduction than to localize Gal4 on the consensus sequence, in turn suggesting that conformational freezing upon docking, rather than the binding strength, might be a key to understanding the biological function of Gal4. Indeed, conformational freezing could be instrumental in the specific binding of Gal4 with several other coactivator proteins, a necessary step toward the activation of the gene expression [42].

### 2.4. Structure and Interface of the DNA–Gal4 Complex: A Molecular Dynamics Study

We thus investigated, through state-of-the-art all-atom molecular dynamics simulations, the conformational freedom and interface of DNA and Gal4, both when isolated and within their complex in the presence or absence of a consensus sequence. We tracked their relative motion and evaluated their thermodynamics in the binding.

Figure 5A–C displays representative snapshots of the isolated Gal4 and of the specific and nonspecific Gal4–DNA complexes, respectively. While the isolated Gal4 assumes a rather compact conformation, as already suggested [21], the interaction with DNA forces more open protein configurations. To quantify such an effect, we calculated the secondary structure percentage along the amino acid sequence, subdivided into an unstructured coil, beta-like configuration or alpha helix. In Figure 6A, we plot the lost and gained secondary structures as the differences between specific binding and isolated proteins (top panel) and between nonspecific binding and isolated proteins (bottom panel). Regions of DNA–Gal4 close proximity are shaded. Most of the interface tracts undergo significant conformational changes, mainly from folded to coil configurations (blue to white bars), while regions not involved in the binding are much less affected. The transition is more marked in the specific complex, such as for residues 15–25.

To address the structural changes of the DNA sequences, we computed several inter-base-pair quantities, the grooves’ depth and width and the bending angle for each frame of MD simulations using Curves+ [39]. In the case of specific DNA sequences, we also observed a 9° increase in the average bending of the DNA toward the protein, as compared to the isolated DNA oligomer (Figure 5B). This significant structural effect, which contributes to the overall conformational entropy loss, can be appreciated in other inter-base-pair quantities (like twist) and in the groove dimensions (Appendix A). On the contrary, in the case of nonspecific DNA sequences, no relevant changes were observed.

The conformational changes are correlated with the relative motion between Gal4 and the DNA, as measured through the center of mass of the protein via the RMSD time series (Figure 6B) or the change of distance along the *z*-axis parallel to the DNA axis (∆z) between a specific amino acid and the base pairs involved in HB (Appendix A). Repeated simulations show that Gal4 in a complex with the nonspecific DNA sequence can visit different binding sites, since the changed ∆z can reach the value of 15 Å, and it remains quite flexible, as demonstrated by the large observed fluctuations. In contrast, Gal4 in complex with the specific sequence retains most of its initial contacts, and much smaller relative movements of the protein along the DNA are observed. The difference between the two situations is also apparent when studying the HBs and their dynamics along the trajectories (Appendix A). Indeed, the average number of active HB at each frame agrees with the crystallographic data and is only slightly larger for the specific case, <n_HB,s_ ≥20 vs. <n_HB,ns_ ≥16. Instead, when considering stable HB along the trajectory (with an occupancy 30% or higher), specific HB are much more, n_HB,s_ = 14 vs. n_HB,ns_ = 6, indicating that the contacts are mostly preserved in the former case, while they are continuously refreshed in the latter. Moreover, for the nonspecific case, in the two repeats, different HB are observed, suggesting that different conformations can be explored, and if the simulations were longer, the breaking of these HB and the changing of the relative position between the protein and the DNA would likely be observed. Overall, these findings, together with the number (147 ± 11 and 153 ± 14 for the specific and nonspecific complexes, respectively) and distribution of interfacial water molecules solvating DNA and protein (Appendix A), defined as water molecules at a distance of 4 A° of both the protein and the DNA sequences, indicate the crucial importance of taking into account the dynamic nature of the interfaces to correctly describe the stability and specificity of Gal4 binding.

### 2.5. Entropy–Enthalpy Compensation upon Binding from the Molecular Dynamics Study

The all-atom molecular dynamics simulations can also lead to estimates of the various contributions to the binding free energy that, albeit difficult to quantitatively compare to experimental results, can support their interpretation. Among them (defined and discussed in Appendix A), we identified polar terms—including electrostatic energy and the polar contribution to the solvation energy and nonpolar terms—including Van der Waals interactions and related to the different number of HB and contacts for the two binding modes. Such terms are all favorable for both binding modes with respect to the unbound state (Appendix A), with an overall difference between specific and nonspecific complex of about 35 kcal mol^−1^, indicating that specific Gal4–DNA binding is strongly favored by enthalpy. This figure is counterbalanced by entropic contributions way more favorable for the nonspecific interactions (*TΔS* ~ –17 kcal mol^−1^) than for specific binding (*TΔS* ~ +26 kcal mol^−1^). Several factors contribute to this entropy difference, as further discussed in Appendix A and Appendix A: (i) the protein has access to a significantly higher number of conformations when bound to the nonspecific DNA sequence (Figure 5), (ii) the protein can slide along the dsDNA only when undocked (Figure 6B) and (iii) the dsDNA is bent and stiffened by Gal4 binding in this specific case (Appendix A). Although additional terms not considered here, like the entropic effect due to the loss of bound water molecules when forming the complex [43] (a large number of water molecules was observed for the specific complex; see Appendix A for more details), may also contribute to the total binding affinity, and despite the potential role of several simplifying assumptions, our simulations unequivocally show a very relevant entropy–enthalpy compensation mechanism for the specific binding in which a relatively small free-energy reduction results from the differences of large quantities.

## 3. Discussion and Conclusions

In this article, we report a quantitative analysis of the binding of the yeast gene regulator Gal4 to dsDNA. We measured the binding strength and kinetics of Gal4 to dsDNA oligomers with different sequences thanks to the real-time multiplexing capacity of the recently introduced RPI technology. The Gal4–DNA interaction is only one of several molecular interactions required for galactose-dependent gene control in yeast, which inevitably limited the scope of our modelling. Nevertheless, the mode of specific DNA recognition by Gal4, a paradigmatic transcriptional activator, is especially worthy of being fully understood, as it has the potential to affect the subsequent events that include the galactose-dependent unmasking of transcription activation domains followed by their interactions with subunits of various coactivator complexes, like SAGA and Mediator [44].

Overall, the combination of our experimental observations, comparative analysis through a simple model and molecular dynamics simulations suggests the following description of the Gal4–DNA recognition process. The protein is attracted towards DNA primarily by an electrostatic interaction, finely modulated by ionic strength. The first binding is not specific to the DNA sequence and involves a relevant number of HBs, although it leaves a large conformational freedom to the protein–DNA complex. Upon random sliding and rearrangements, the protein binds to the consensus sequence with an enthalpy gain almost compensated by a large entropy loss.

Our quantitative analysis conveys new clues to understand the specificity and efficacy of the action of Gal4, which may be relevant for various other transcription factors. It has been repeatedly noticed that the specificity of transcription factors, especially in yeast and eukaryotic cells, is not sufficient to provide the necessary transcription selectivity, which can only be provided by cooperativity among the different transcription factors at the same DNA regulatory regions. Indeed, our results on Gal4 imply that, when the target sequence is diluted in the 10^7^ base-long yeast genome, the nonspecific binding should largely dominate, even assuming a large fraction of chromatinization. While this might appear the reasonable and expected condition enabling the search for the needle-in-the-haystack cognate site through sliding and hopping, one might also wonder what prevents spurious transcription signaling. We argue that the large entropic cost in specific binding, associated to a precisely shaped Gal4–DNA complex, might be a critical element in recruiting the cofactors necessary to initiate the transcription, thereby minimizing the ectopic transcription initiation events. Accordingly, out of the many conformations that are compatible with interacting with a generic dsDNA, only a tiny fraction matches those induced by specific docking. The overall outcome of these effects might well be described as DNA sequence-induced structural changes, as those reported for glucocorticoid receptor binding to its cognate site [45,46]. We speculate that such conformational constraints, combined with the docking lifetime, are key to the biological action of Gal4, as well as of other TFs.

## 4. Materials and Methods

### 4.1. Biomolecules and Reagents

We studied a stable recombinant *Saccharomyces cerevisiae* Gal4 N-terminal fragment, comprising amino acids 1–147 (MW = 16.9 kDa), purchased from Abcam (Abcam, Cambridge, UK). Gal4(1–147) specifically binds DNA as a dimer. It dimerizes in the solution in the absence of DNA [47]. Amine-terminated oligonucleotides (Ultramers, Integrated DNA Technologies, Coralville, IA, USA) were spotted on the RPI sensor surface, as detailed in Appendix A. Gal4 was suspended before use in the measuring buffer (Tris HCl, pH 7.5, 50 mM, Tween 20 0.02%, NaN_3_ 0.02%, ZnSO_4_ 200 μM and NaCl from 50 to 250 mM).

### 4.2. RPI Measurements

The RPI measurements were performed by using the experimental set-up and the analysis procedure described in reference [18]. Briefly, Gal4 was injected into the RPI cartridge to reach a final concentration *c* from 0.08 nM up to 50 nM. We avoided larger protein concentrations that can result in aggregation. All the experiments were performed at 30 °C under stirring. Raw images of the reflecting surface were converted into surface density signals, as detailed in Appendix A. The binding curves were analyzed as described in the text to extract the equilibrium and kinetic parameters.

### 4.3. All-Atom Molecular Dynamics Simulations and Analysis

To model Gal4, we started from the crystallographic structure of the DNA-bound protein (PDB 3COQ). For DNA sequences, the free structure was energy-optimized using the internal/helicoidal variable modeling JUMNA [48] with the AMBER par98 force field. For the nonspecific complex, we superimposed the specific complex and the nonspecific DNA sequence to determine the corresponding protein–DNA contacts that have to be preserved during the minimization of the protein–DNA nonspecific complex. For the protocols, solvent models and Debye–Hückel salt treatment, see Appendix A. Microsecond all-atom molecular dynamic (MD) simulations were performed using the GROMACS 5 package [49] on the protein and DNA molecules alone and on their specific and nonspecific complexes (two repeats). See Appendix A for all details on the MD protocols and HB treatments. The conformational analysis of the dsDNA was performed using Curves+ [50], which provides a full set of helical, backbone and groove geometry parameters. The HB and solvating water molecules were identified based on the distance and angle cut-offs (see Appendix A) upon which the thermodynamic quantities were estimated.

## Figures and Tables

**Figure 1 ijms-22-03813-f001:**
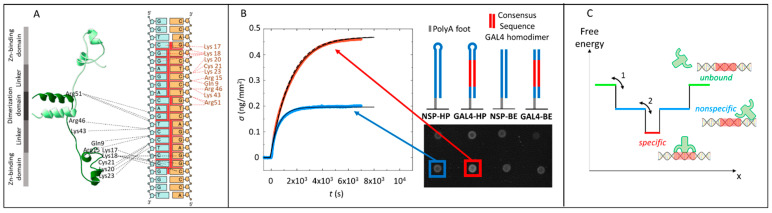
Gal4–DNA interaction measured by the Reflective Phantom Interface (RPI). (**A**) Sketch of the Gal4 (8–64)/DNA complex based on The Protein Data Bank (PDB) 3COQ, with the two subunits of the dimer colored in light and dark green, respectively. Dashed lines indicate hydrogen bonds (HB) between Gal4 amino acids and the consensus sequence. (**B**) Binding curves of Gal4 on specific and nonspecific DNA sequences measured by the RPI. Time evolution of the surface density of protein binding on two different spots with specific (red) and nonspecific (blue) DNA sequences upon the addition of 10-nM Gal4 in the solution. Sketches of the immobilized sequences and RPI images of the DNA spots are reproduced aside. Black lines are fit to the experimental curves with Equation (4). The dashed line corresponds to the highest calculated rate of binding, as discussed in Appendix A. (**C**) Sketch of the energy profile of the protein–DNA complex vs. a generic reaction coordinate x. The width along x of the nonspecific and specific wells mimic the number of conformations accessible to the system.

**Figure 2 ijms-22-03813-f002:**
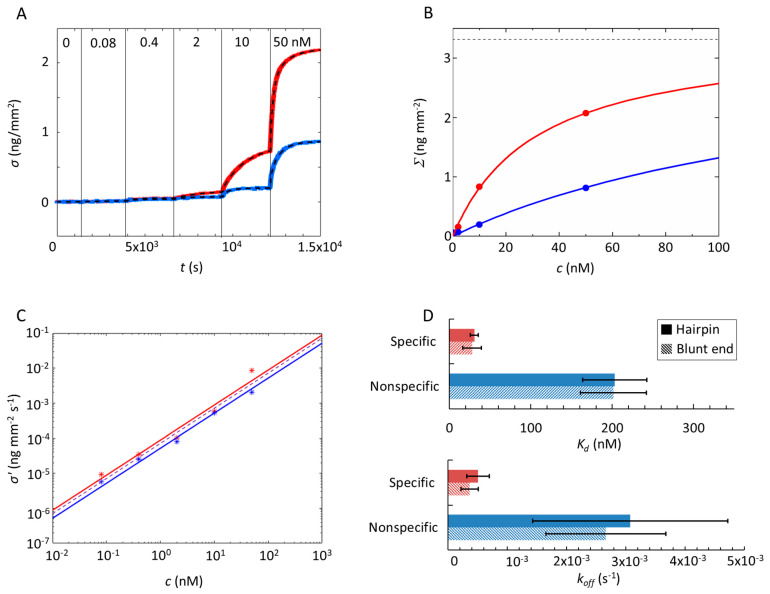
Analysis of the Gal4–DNA interaction. (**A**) Increase of the surface density (*σ*) measured over time upon protein binding to specific (red) and nonspecific (blue) sequences for increasing the concentration of Gal4 in a solution (ionic strength *I*_s_ = 150 mM, T = 30 °C and *c* = 10 μM). At the time indicated by the vertical lines, the Gal4 concentration was increased stepwise from 0 to 50 nM, as reported in the figure. The dashed curve superimposed to the data represents the fitting with exponential growth functions (Equation (1)), from which the rate *Γ*(*c*) and the amplitude *Σ*(*c*) of the binding curves are extracted. (**B**) The amplitude *Σ*(c) for specific and nonspecific probes (same colors as in panel A) upon an increase of protein concentration. Lines are fits with the Langmuir model Equation (2). The dashed horizontal line represents the asymptotic value for *Σ*(*c*→*∞*) obtained from the fit. (**C**) Initial slope *σ*’(*c*) of the exponential fit shown in (A) for specific and nonspecific probes. Continuous and dashed lines are separate and common fits to the data, respectively, as discussed in the text. (**D**) Bar graph showing the equilibrium dissociation constant (*K_d_*) (upper) and *k_off_* (lower) of Gal4 on double-stranded DNA (dsDNA) (with hairpin or blunt ends) for specific and nonspecific sequences. The error bars represent the SD computed from three independent experiments.

**Figure 3 ijms-22-03813-f003:**
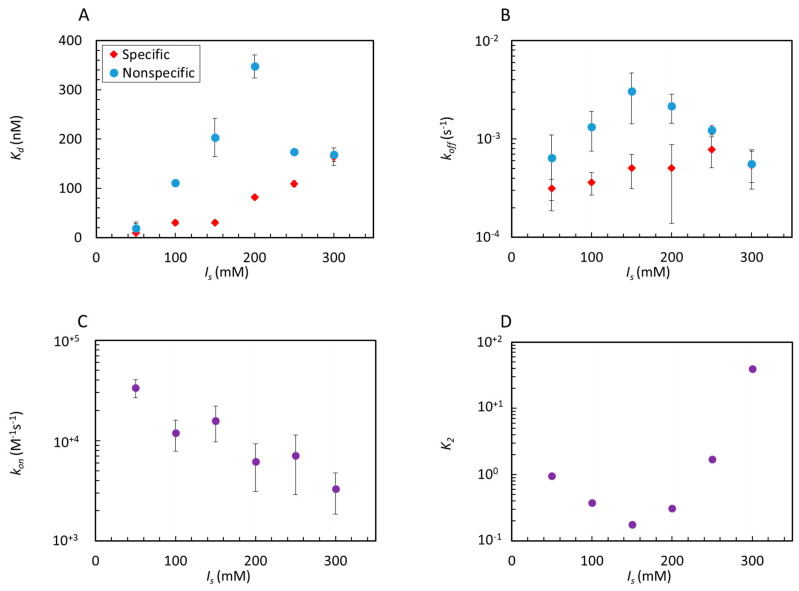
Effect of ionic strength on Gal4–DNA interactions. Equilibrium dissociation constants *K_d_* (**A**) and dissociation rates *k_off_* (**B**) for specific (red) and nonspecific (blue) hairpin sequences. (**C**) Association rate *k_on_*, common to specific and nonspecific strands (purple). (**D**) Estimated specific dissociation constant *K_2_* from Equation (3).

**Figure 4 ijms-22-03813-f004:**
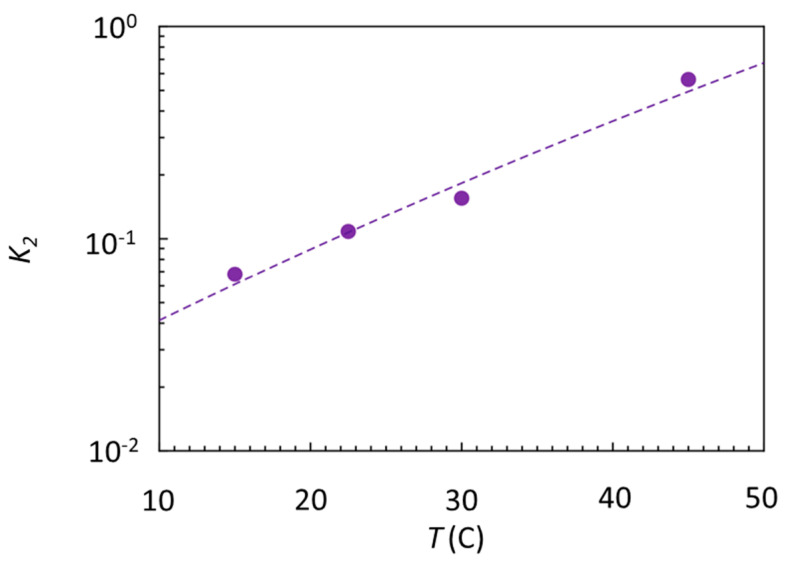
Effect of temperature on the Gal4–DNA interaction. Estimated specific equilibrium dissociation coefficient *K*_2_ (dots) obtained from the ratio of the dissociation constants measured on spots of specific and nonspecific strands according to Equation (3).

**Figure 5 ijms-22-03813-f005:**
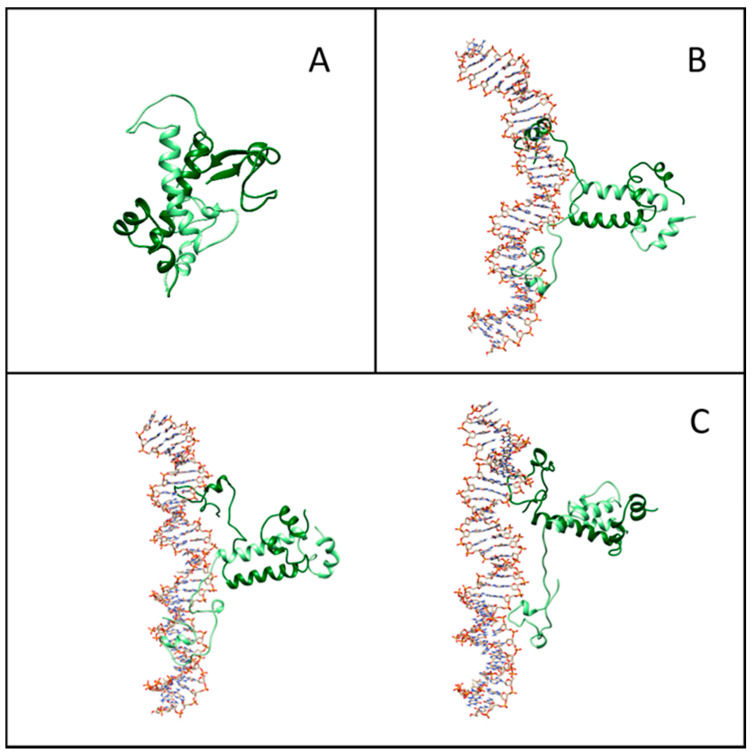
Representative structures from molecular simulations. (**A**) Equilibrated structure of isolated Gal4. (**B**) Snapshot of the structure of the Gal4–DNA complex in the presence of the specific DNA sequences. (**C**) Examples of structures of the Gal4–DNA complex in the absence of the specific DNA sequences.

**Figure 6 ijms-22-03813-f006:**
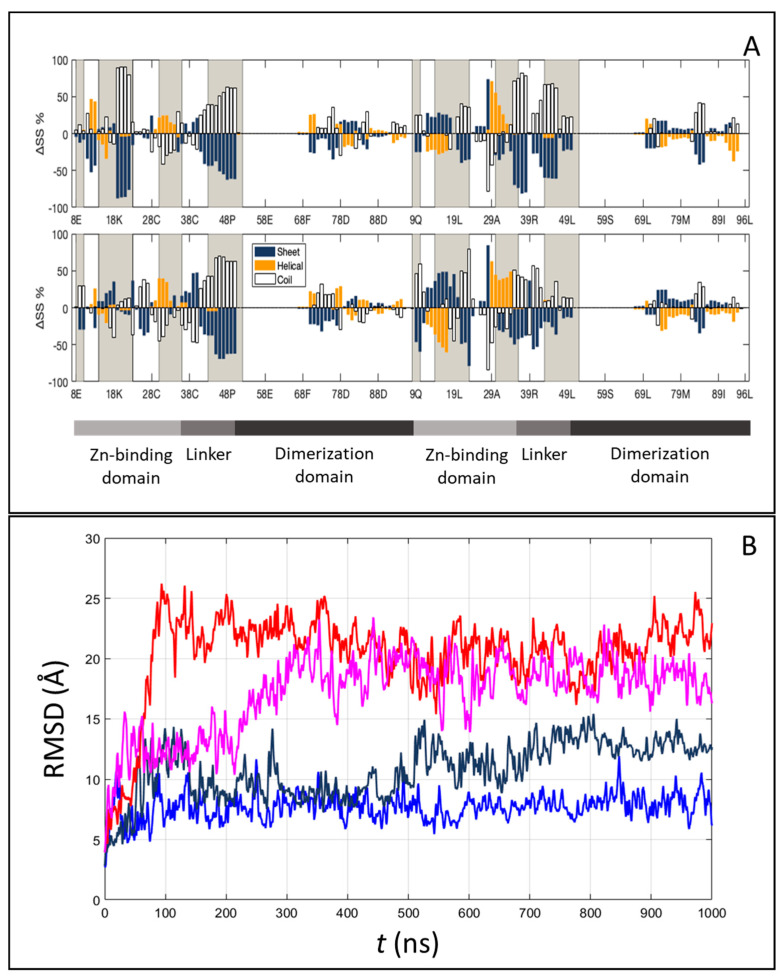
Conformational and translational freedom within the Gal4–DNA complex from molecular simulations. (**A**) Difference in the average protein secondary structure percentage calculated along the molecular dynamic (MD) simulations, between the specific complex and the isolated protein (top) or the nonspecific complex and the isolated protein (bottom). Gained and lost secondary structures are expressed as positive and negative bars, respectively. The grey regions represent the interface portions in the protein. (**B**) Protein movements along the DNA are computed as the Root Mean Square Deviation (RMSD) of the protein after the superposition of DNA heavy atoms, with specific (blue and dark blue) and nonspecific (red and magenta) sequences.

**Table 1 ijms-22-03813-t001:** DNA oligomers used in the study. Sequences 1-4 were grafted on the Reflective Phantom Interface (RPI)-sensing surface. Sequences 5 and 6 were used to hybridize sequences 3 and 4, respectively. The red part represents the region that differs between specific and nonspecific strands. CGG and CCG sequences, important for GAl4 binding, are underlined.

#	Name	Length	Sequence
1	GAL4-HP	106	/5AmMC6/AAA AAA AAA ATG AAA TGT TGG AAG GGT CGG AGG ACA GTC CTC CGG GTG GTA TAG TCT CCT ACC TAT ACC ACC CGG AGG ACT GTC CTC CGA CCC TTC CAA CAT TTC A
2	NSP-HP	106	/5AmMC6/AAA AAA AAA ATG AAA TGT TGG TTG CGT CTC TCC TAT GTT GCG TCG GTG GTA TAG TCT CCT ACC TAT ACC ACC GAC GCA ACA TAG GAG AGA CGC AAC CAA CAT TTC A
3	GAL4-BE	54	/5AmMC6/AAA AAA AAA ATG AAA TGT TGG AAG GGT CGG AGG ACA GTC CTC CGG GTG GTA TAG
4	NSP-BE	54	/5AmMC6/AAA AAA AAA ATG AAA TGT TGG TTG CGT CTC TCC TAT GTT GCG TCG GTG GTA TAG
5	GAL4-BE-C	44	CTA TAC CAC CCG GAG GAC TGT CCT CCG ACC CTT CCA ACA TTT CA
6	NSP-BE-C	44	CTA TAC CAC CGA CGC AAC ATA GGA GAG ACG CAA CCA ACA TTT CA

## Data Availability

All data are contained within this article and the associated supporting information.

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
