# Peer review of "A Bit Stickier, a Bit Slower, a Lot Stiffer: Specific vs. Nonspecific Binding of Gal4 to DNA"

_ijms, 2021, doi:10.3390/ijms22083813_

Round 1

Reviewer 1 Report

Authors of this manuscript have designed the study and executed of the experiments well.

Well written manuscript. RPI is very interesting label free optical technology to understand the biomolecular specific/non-specific interactions. Minor concern- As nucleus is highly crowded environment, it would be a complete study if you perform same experiments in presence of molecular crowding agents (eg., different size of PEGs).

Reviewer 2 Report

This manuscripts describes results from study of Gal4 binding to DNA, both a consensus binding site sequence and other non-consensus sequences, using RPI and also from MD simulations. The measurement method allows both binding kinetics and equilibrium binding constants to be determined, which is valuable. From their interpretation of the results a surprising finding is that there is only a small difference in binding free energy to specific and non-specific sequences. The MD simulations and thermodynamic partitioning of contributions to deltaG indicate major difference in entropy for the complex bound at specific and non-specific sites, and the authors suggest this may have a role in function though this idea is not really developed into a clear model for this. 

While the results are potentially interesting, the presentation is problematic. It seems to this reviewer that the authors have tried to pack too much into one manuscript, and the presentation suffers from not fully and clearly developing ideas in the body of the manuscript. There is the measurement methodology (spotting concentration, background correction, etc.), developing a kinetic model and justifying its use, description and discussion of the MD simulation results, and combining to give a handwaving thermodynamic model that is not well tied to function. 

There are aspects of the experiment design that need clarification:

There is no mention of the fact that binding in this experiment has a lower dimensionality (as does SPR) than a bulk solution measurement, any potential effect should be discussed. 

While the consensus CGG contribute most to the binding, a consensus does not always binding very tightly. The discussion of the choice of sequence (in S.I.) was not very good. 

The way sequences are presented in Table 1 is poor for comparison of the oligos used. 

The ‘specific’ oligomers probably have one tight binding site, but for ‘non-specific’ what is the effective number of sites (and hence concentration)? How does the multiplicity of possible sites affect the analysis?  

It was quite surprising to see that the single stranded DNA control has essentially the same affinity as the non-specific duplexes - can this be explained? 

There are also issues with language. For example, page 7 refers to activation of hydrogen bonds (does this mean formation of them?), and consensus interactions that are pre-empted by those to non-specific DNA - it is really not clear what this is intended to mean. The notation in the kinetic model is clumsy. In discussion of the MD simulations reference is made to secondary structure propensity. SS propensity is generally used as a term for predictions of the tendency of a segment of a protein to make that secondary structure, where the prediction is made from the amino acid sequence alone. In this work the SS propensity seems to mean the percentage of time an amino acid is found to form that secondary structure during the simulation - the switch from the standard meaning is confusing. Page 9 what does symmetrically large mean? 
